# Olfactometer Responses of Convergent Lady Beetles *Hippodamia convergens* (Coleoptera: Coccinellidae) to Odor Cues from Aphid-Infested Cotton Plants Treated with Plant-Associated Fungi

**DOI:** 10.3390/insects13020157

**Published:** 2022-01-31

**Authors:** Janaina Camara Siqueira da Cunha, Morgan H. Swoboda, Gregory A. Sword

**Affiliations:** 1Department of Entomology, Texas A&M University, College Station, TX 77843, USA; mhs338@cornell.edu; 2Department of Entomology, Cornell University, Ithaca, NY 14850, USA

**Keywords:** olfactory preference, multitrophic interactions, microorganisms, predator

## Abstract

**Simple Summary:**

The cotton aphid *Aphis gossypii* is a serious agricultural pest. Microbes associated with plants can affect the behavior and performance of insect herbivores and their natural enemies. *Phialemonium inflatum* and *Chaetomium globosum* fungi can reduce cotton aphid reproduction when applied as a seed treatment to cotton. We evaluated whether these fungi might affect the interaction between cotton aphids and a natural enemy, the convergent lady beetle *Hippodamia convergens*. We used dual-choice olfactometer experiments to assess lady beetle behavioral responses to cues from fungal-treated cotton plants in the presence or absence of aphid infestations. In the absence of fungal treatments, males preferred odors from aphid-infested relative to non-infested plants, and females spent more time associated with olfactory stimuli from aphid-infested versus non-infested plants. When cues from fungal-treated plants infested with aphids were assessed, there were no differences in lady beetle responses. The only fungal treatment-related effects involved plants without aphids. In the absence of aphids, males responded slower to *P. inflatum*-treated plants compared to control, and females preferred *P. inflatum*-treated plants. Treating cotton with these potentially beneficial fungi had minor effects on lady beetle behavioral responses and would not be expected to disrupt this predator–prey–plant interaction as part of an integrated pest management strategy.

**Abstract:**

Microbes have the potential to affect multitrophic plant–insect–predator interactions. We examined whether cotton plants treated with potentially beneficial fungi affect interactions between cotton aphids *Aphis gossypii* and predatory lady beetles *Hippodamia convergens*. We used Y-tube olfactometer assays to test lady beetle behavioral responses to stimuli emitted by aphid-infested and non-infested cotton plants grown from seeds treated with either *Phialemonium inflatum* (TAMU490) or *Chaetomium globosum* (TAMU520) versus untreated control plants. We tested a total of 960 lady beetles (480 males and 480 females) that had been deprived of food for approximately 24 h. In the absence of any fungal treatments, males preferred stimuli from aphid-infested plants, and females spent more time associated with stimuli from aphid-infested versus non-infested plants. When fungal treatments were added, we observed that lady beetles preferred non-aphid-infested *P. inflatum* plants, and males responded slower to plants treated with *P. inflatum* in the absence of aphids. We found some evidence to suggest that lady beetle behavioral responses to plants might vary according to the fungal treatment but not strongly impact their use as part of an insect pest management strategy.

## 1. Introduction

A phytobiome is the association between plants, the environment, and micro- and macroscopic organisms influencing plant growth, health, and productivity [1,2]. A wide variety of studies have shown that plant-associated microbes, including fungi, can enhance plant resistance or tolerance against biotic and abiotic stressors such as insect herbivores, pathogens, plant-parasitic nematodes, drought, and heat [3,4,5,6,7,8,9,10,11,12,13].

Many plant-associated microbes can induce plant host defenses through Systemic Acquired Resistance (SAR) and Induced Systemic Resistance (ISR) [4,14,15]. For example, a laboratory study inoculating maize seeds with an endophytic fungus showed the fungus promoted plant growth, altered the expression of defensive genes belonging to the jasmonic acid (JA) pathway, and suppressed herbivore larvae growth rate [16]. Moreover, microbes can affect the production of various chemicals by the plant, including volatile organic compounds (VOCs), thereby modifying plant responses [17,18,19]. These altered volatile profiles can affect herbivore host-selection behavior [20,21,22]. Consequently, these changes in volatile chemical bouquets could also affect the attraction of natural enemies, such as predators and parasitoids [23,24].

The cotton aphid, *Aphis gossypii* Glover (Hemiptera: Aphididae), is a well-known pest that can cause severe economic losses in cotton fields [25,26]. Some plant-associated fungi have been shown to negatively affect cotton aphid reproduction and alter feeding behavior [5,8,27,28]. Diverse species in the family Coccinellidae (Coleoptera) are voracious aphid predators [29] and agriculturally valuable biological control agents [30,31]. One example is the generalist predator *Hippodamia convergens* (Guérin-Méneville), commonly known as the convergent lady beetle. This aphidophagous species is found broadly across the Western Hemisphere [32]. Due to their predaceous habit and distribution among crops attacked by aphids, this species is often considered an important part of many agroecosystems as an essential biological control agent [33]. However, most studies assessing the effects of plant-associated fungi on lady beetles have only been limited to grasses [34,35,36] despite its presence in many other crops, including cotton [37,38].

In order to better manage insect pests, we may be able to manipulate a plant’s phytobiome to increase the efficiency of natural enemies [39,40]. Knowledge about pests and their natural enemies is crucial for the development and implementation of sustainable pest management strategies in cotton [39]. Cotton plants treated with some beneficial plant-associated fungi have previously been shown to negatively affect aphid reproduction [5,8]. However, whether these plant-associated fungi might also affect the behavior of an aphid predator in a multitrophic interaction has not been investigated to date. As such, the goal of this study was to investigate the effects of plant-associated fungi applied to cotton plants on convergent lady beetle behavior.

## 2. Materials and Methods

### 2.1. Fungal Treatment of Cotton Seeds

Chemically untreated *Gossypium hirsutum* seeds of the non-transgenic variety LA122 were obtained from All-Tex Seed Inc., Levelland, TX, USA. The fungal strains used were *Phialemonium inflatum* (TAMU490) and *Chaetomium globosum* (TAMU520), which were first isolated as endophytes from surface-sterilized cultivated cotton as part of a field survey in Texas, USA [41]. The fungal inoculum for all trials was cultured in 100 × 15 mm Petri dishes on potato dextrose agar (PDA) in the dark at 25 °C. Spore suspensions of each fungus were made by adding 2 mL of 0.1% Triton X-100 solution to the fungal conidia plates, scraping them with a sterile metal spatula, filtering through autoclaved 0.25 mm sieves into a sterile beaker, and placing them in 50 mL centrifuge tubes [9]. The suspensions were mixed on a vortex and then centrifuged for 10 min in a Cole-Parmer fixed-speed centrifuge at 3000 rpm. Excess water was removed by pouring out the supernatant. We used a Neubauer hemocytometer (Thomas Scientific, Philadelphia, PA, USA) to quantify the spores’ concentration. Final treatment concentrations were diluted with sterile water to reach 1 × 10^8^ spores/mL [9].

Cotton seeds were surface sterilized by immersion in 3% sodium hypochlorite (NaOCl) for 3 min, and 70% ethanol for 2 min, followed by three rinses in sterile water [42]. Before applying the fungal treatment, surface-sterilized seeds were dried on sterile paper towels for 30 min. The seeds were inoculated with spore suspensions (approximately 200 seeds/1 mL) plus 1 mL of a 2% methylcellulose sticker to bind the spores to the seeds. We treated the control seeds with 1 mL of 2% methylcellulose only. Treated seeds were dried for at least three hours after inoculation before planting. Five treated seeds per treatment were plated in Petri dishes containing PDA to confirm inoculation with viable fungi [9]. Three seeds per treatment were planted in 515 mL pots with unsterilized MetroMix 900 soil (Sun Gro Horticulture, Agawam, MA, USA). For the duration of the experiment, all plants were grown in a greenhouse at approx. 25 °C with natural photoperiod. Pots were randomized and watered as needed at least once a week.

### 2.2. Insect Rearing and Experimental Design

We assessed lady beetle behavioral responses to olfactory stimuli emitted by cotton plants grown in the USA from seeds treated with plant-associated fungi using a dual-choice Y-tube olfactometer (described below). To prepare aphid-infested plants for the behavioral assays, third true-leaf plants from each fungus and untreated control treatments were initially infested with 10 aphids per plant two weeks before the trials with cotton aphids from a colony maintained in the Sword Lab at Texas A&M University. The colony was maintained on cotton plants in the greenhouse at 25 °C with natural photoperiod ranging from 12L:12D to 14L:10D, with new non-treated plants being placed in the cages weekly. A total of 18 infested plants per treatment were maintained inside multiple insect mesh cages and housed in a greenhouse under the conditions mentioned above. Plants from all three treatment groups that were not infested with cotton aphids were maintained in the same environmental conditions as the infested plants. Although previous studies have shown a decrease in aphid reproduction on plants grown in the USA from fungal-treated seeds [5,8,11], aphid populations on treated plants nevertheless increase over time. In these experiments, aphid populations on treated plants had recovered by two weeks after infestation such that all plants had similar aphid infestation levels.

Convergent lady beetle adults, *Hippodamia convergens*, were obtained from ARBICO Organics^®^ (Oro Valley, AZ, USA) originally collected from overwintering aggregations in California, USA [43,44]. Prior to use in the trials, the beetles were sexed and maintained in reproductive diapause in 44 mL plastic cups at 3 °C [45,46]. For use in experiments, ten individuals per cup were arranged randomly on trays inside an incubator at 25 °C, 50–60% RH, and 16:8 L:D photoperiod [44]. The lady beetles were fed once for 24 h with approx. 30 cotton aphids per adult from the aphid colony and a moistened cotton wick inside the cups in the incubator before starting the pre-experiment starvation period. Convergent lady beetles were also field collected from sorghum plants at the Texas A&M AgriLife Research Farm in Burleson County, TX and maintained under the same conditions. All adult lady beetles were starved ~24 h before the behavior assays [47,48,49].

Because the lady beetles came from two different source populations as described above, we conducted a control experiment to test for differences between the commercial and wild-caught individuals in their preference for stimuli from cotton plants with and without aphids in the absence of any fungal treatments. No difference was observed (see Results). We then tested for the effects of fungal cotton seed treatments on lady beetle behavior by conducting choice tests between stimuli from one untreated control versus one fungal-treated plant (TAMU490 or TAMU 520). We conducted this comparison between untreated and fungal-treated plants using plants that were either aphid-free or aphid-infested in two separate series of trials. We used a total of 960 adults, 120 for the initial comparison between commercial and wild-caught populations (60 males and 60 females per population), 120 for the untreated control aphid-infested and non-infested plants, 240 in each comparison of fungal-treated plants aphid-infested and non-infested plants, and 240 in each of the two separate comparisons between fungal-treated and untreated plants in either the presence or absence of aphids. Plant positions (left or right) in the olfactometer were alternated after every five individuals, and new plants were exchanged after every 30 individuals. Each adult was tested only once and discarded after the experiment.

### 2.3. Y-Tube Olfactometer

The olfactometer consisted of a Y-shaped glass tube with a trunk measuring 15.2 cm and each arm 12.7 cm (Figure 1). Two 2 L glass jars were attached to the outside as chambers for each plant. The jars’ lid was modified with an opening through which the plant stem was placed. The space between the lid and stem was then sealed with non-toxic clay to avoid air escaping, thereby isolating the stem and leaves within the jar. A filter was connected in series to a water bubbler to humidify the incoming air pulled from a DOA series oilless diaphragm vacuum pump (Thomas Scientific). The filters were attached to silicone tubes, and the flow was measured with an Acrylic Flowmeter (Cole-Parmer Scientific Experts, Illinois, USA). The olfactometer was positioned horizontally on a countertop [50,51] inside a dark room. The light source came from a flexible LED strip light equidistantly placed to provide uniform light to both arms of the olfactometer. Carbon filtered humidified air was pumped in at ~2.0 L min^−1,^ and a single adult convergent lady beetle was introduced at the base of the Y-tube olfactometer. The air was monitored, checking the flowmeter during the whole observation to ensure it was not escaping and interfering with the assay.

After five individuals were tested, we changed the Y-tube, the jars, and the treatment sides to avoid positional bias [52,53]. Jars were cleaned with fragrance-free soap, rinsed with water, and dried in an oven at 80 °C to sterilize and avoid residuals from the previous treatment [50]. Adult lady beetles were gently introduced into the release chamber with a #2/0 Daler-Rowney paintbrush and allowed to acclimate for five minutes [50]. Consistent with previous olfactometer studies, the insect had 10 min to choose between the different stimuli [54,55,56]. We recorded the insect responses as a choice when they entered at least halfway up into one arm of the Y-tube and remained there for at least 20 s [51]. Within the 10 min, we recorded the first choice, latency (time to make a choice), and residence time (time spent in an arm) [52,53]. If an individual did not choose within five minutes, it was recorded as “no choice” and excluded from the statistical analysis [57,58,59].

### 2.4. Statistical Analysis

We recorded the number of responding lady beetles (females and males) and expressed it as a proportion calculated as the number of individuals that chose a given treatment divided by the total number of individuals that selected either the treatment or control stimulus. The proportions of responding individuals yield a value between 0 and 1 [9,60]. We analyzed the proportions using Pearson’s chi-squared test [61], testing the null hypothesis that *H. convergens* showed no preference for either arm, and the expected proportion was equal to 0.5 [59,62]. The latency and residence time data were transformed to satisfy the assumptions of normality using log (x + 1) [63] and compared the means of each sex between treatments using Welch’s two-sample *t*-test [52,61,64]. We used ANOVA to analyze if there was a difference between wild and commercial lady beetle responses. All analyses were done using R version 3.6.3 [61] with a 5% significance level (α = 0.05), and we used the ggplot2 package for graphs [65].

## 3. Results

### 3.1. Wild and Commercial *H. convergens* Responses

We found no differences in the behavior of wild versus commercially obtained lady beetles in response to aphid-infested and non-infested cotton plants in the absence of any fungal treatments. The first-choice responses from *H. convergens* wild and commercial females and males were not significantly different (*F*_1,91_ = 0.0141, *p* = 0.9132). Moreover, there was no significant difference in either latency (*F*_3,89_ = 2.01, *p* = 0.1182) or residence time (*F*_4,88_ = 0.3669, *p* = 0.777) between wild and commercial individuals of both sexes. Since we did not find a significant difference between responses of wild and commercial *H. convergens*, their responses were not incorporated in the analysis.

### 3.2. First Choice

In the Y-tube olfactometer, *H. convergens* females did not show a significant preference for stimuli emitted by untreated cotton plants that were either infested or not infested with aphids. However, when the females were exposed to stimuli from *P. inflatum* fungal-treated plants with or without aphids, they significantly preferred stimuli from non-infested plants more often. There were no significant differences in the first choices between stimuli from plants that had been treated with either fungus versus untreated control plants, regardless of whether the plants were infested or not with aphids (Table 1, Figure 2). In contrast, *H. convergens* males did show a significant preference for aphid-infested plants over non-infested plants in the absence of any fungal treatments, but the fungal treatments did not affect their responses regardless of whether aphids were present or absent (Table 1, Figure 2).

### 3.3. Latency to First-Choice

For the *H. convergens* females, no significant differences were found in the latency to their first choice among any treatment pairs (Table 1, Figure 3). However, the males exhibited a significant difference in the absence of aphids in cotton plants treated with *P. inflatum*, taking more time to choose the stimuli emitted by the fungal-treated plants relative to untreated control plants. No other significant differences in latency to the first choice were observed in any other treatment comparisons (Table 1, Figure 3).

### 3.4. Residence Time

In the absence of any fungal treatments, *H. convergens* females spent more time in association with the stimuli emitted by aphid-infested plants, whereas the response of males was not significantly different. Fungal treatments had no effect on the residence time of the insects in response to stimuli from aphid-infested versus non-infested plants, nor were there any differences between fungal-treated and untreated plants, regardless of whether aphids were present or absent (Table 1, Figure 4).

## 4. Discussion

Commonly, aphid-damaged plants are more attractive to lady beetles than non-infested plants [66,67]. *Hippodamia convergens* have previously been shown to be strongly attracted to the odor emitted by plants infested with aphids [68] and to the aphid alarm pheromone [69]. Other predatory beetle species have shown a similar preference for plants infested with aphids as well, such as *Coleomegilla maculata* (De Geer) females that significantly preferred fava bean plants infested with pea aphids [55]. Moreover, some piercing-sucking insects, including aphids, induce salicylic acid (SA) signaling mediated by feeding, triggering systemic acquired resistance (SAR) in the plant. SAR is primarily thought of as a defense against plant pathogens and can involve plant volatiles [70,71,72]. Thus, herbivore-induced plant volatiles (HIPVs) and semiochemicals from aphids are potentially available in the environment as olfactory cues for predators and have been shown to affect electroantennogram activity, foraging behavior, and attractiveness of prey to coccinellids [69,73,74,75,76].

For our first hypothesis, we expected *H. convergens* females and males would prefer olfactory stimuli from aphid-infested cotton plants in the absence of any fungal treatments. We partially supported this hypothesis because only the males showed a clear first choice for aphid-infested plants. The females did not exhibit a first-choice preference in these trials. We expected that both males and females would have lower latency times and higher residence times associated with stimuli from aphid-infested plants in the absence of fungal treatments. However, we did not find a significant difference in latency for either males or females. For residence time, the females spent more time associated with stimuli emitted by aphid-infested plants, but there was no effect on male residence time.

*H. convergens* is a generalist predator, but aphids are its primary food source, and the presence of cotton aphids can increase convergent lady beetle feeding and egg viability [37,66]. Thus, we expected that both males and females would have a higher residence time associated with stimuli from aphid-infested plants, but only females showed this pattern in untreated control plants. *Coccinella septempunctata* (Coleoptera: Coccinellidae) females showed a higher attraction to aphid-infested plants [67], which could explain the higher residence time for females. Elliott, Kieckhefer, and Phoofolo [68] found that the high density of aphids influenced the foraging behavior of the convergent lady beetle with increased residence time in both females and males. The possible explanation for this attractiveness was the influence of chemicals (e.g., volatile sesquiterpenes and alkaloids) on prey and habitat location [69,70]. Both alarm and sexual pheromones of aphids act as attractants for the Asian lady beetle *Harmonia axyridis* (Pallas) (Coleoptera: Coccinellidae), indicating that these components influence the beetles’ behavioral responses [71]. Thus, the preference for aphid-infested plants could be related to the difference in the volatile blends from damaged plants.

For the second hypothesis, we predicted that the fungal treatment of cotton plants would affect the beetles’ behavioral responses. The only two significant behavioral responses to fungal treatments observed in these assays involved *P. inflatum*-treated plants. First, females initially chose stimuli from *P. inflatum* treated plants that were not infested with aphids over those that were infested. Secondly, in the absence of any aphids, males took longer to respond to stimuli from *P. inflatum*-treated versus untreated control plants. For the first choice, neither females nor males exhibited a preference for stimuli from untreated control cotton plants versus those treated with either *C. globosum* or *P. inflatum* regardless of whether aphids were present or absent. We also did not find any significant differences in the residence times of either males or females in the presence or absence of aphids with fungal treatments.

We initially predicted the lady beetles would prefer fungal-treated plants if the VOCs emitted by these plants acted as a cue for host finding that increased plant attractiveness to natural enemies [72]. Plant-associated fungi have previously been shown to cause plants to emit different volatile organic compounds (VOCs) profiles, that by attracting natural enemies, can indirectly act as a plant defense mechanism [73,74,75,76]. Other predators, such as *Chrysoperla carnea* (Stephens) (Neuroptera: Chrysopidae), preferred feeding on cotton aphids when plants had been treated with fungi [76]. One possibility for the observation of males taking more time to respond to stimuli from fungal-treated plants in the absence of aphids is that the stimuli from the fungal-treated plants repelled them. The well-known fungus–plant complex *Neotyphodium lolii* and *Lolium perenne* showed negative effects on the aphidophagous *C. septempunctata* fed on cereal aphids, extending larval development, reducing survival and adult fecundity, and reducing reproductive performance [34] that could lead to a repellency behavior in the presence of fungal-treated plants. In the study conducted with the *Neotyphodium*–Arizona fescue complex and the bird cherry-oat aphids, *C. septempunctata* avoided feeding on aphids from plant hybrids with endophyte, showing a preference for other treatments [36]. However, the lady beetles in our experiment had no prior experience with aphids fed on fungal-treated plants, so this seems unlikely as an explanation for the increased latency of males to respond to fungal-treated plants that we observed.

The lady beetle females tended to first select stimuli from *P. inflatum*-treated plants without aphids over stimuli from aphid-infested plants treated with the same fungus. Although we do not know the mechanism underlying this response in our experiments, some herbivores can use secondary metabolites resulting from microbe–plant associations to defend themselves against natural enemies [77,78], making them avoid these plants. In parasitoids, some fungal endophytes can alter the plant alkaloids produced, affecting herbivore susceptibility to natural enemies [79]. Some secondary parasitoids can be negatively influenced by endophytes reducing their lifespan, with experienced females learning to avoid hosts arising from the endophyte–aphid–primary parasitoid interaction [80].

If the fungi played a major role in natural enemy attraction, we would have expected more robust behavioral responses in *H. convergens* towards stimuli from fungal-treated plants, but we did not observe any strong evidence for this in our results. The only observed response to fungal treatment that might have implications for biological control was when plants were treated with *P. inflatum*, and females first preferred stimuli from plants without aphids over those that were infested. If this same response occurred under field conditions, we might expect some reduced lady beetle predation of aphids on *P. inflatum*-treated plants. However, this same strain of fungi has previously been shown to reduce aphid population growth on cotton when applied in the same manner (5). Thus, although the potential exists for a negative trade-off of *P. inflatum* treatment in terms of predation, its impact on population dynamics in the field would be expected to be moderated by the direct negative effects of the same fungal treatment on aphid reproduction. Notably, the same potential for a negative trade-off was not suggested in any of the trials involving *C. globosum* treatment, highlighting the taxonomic specificity in effects on the next trophic level, and the need for more studies investigating the ecological consequences of fungal treatments as an aphid control strategy in the field.

To the best of our knowledge, this is the first study assessing the effects of plant-associated fungi on the behavioral responses of convergent lady beetles. Despite the relatively minor effects observed across the experiments, we did observe some *H. convergens* responses associated with both aphid infestation and fungal treatments of cotton plants. However, it is critical to acknowledge that this laboratory study is a vast simplification of the complex stimuli and their interactions that would occur in the field. For future work, these effects should be assessed under field conditions to determine whether the attractiveness patterns observed here are different at the spatial scales found in agricultural ecosystems. Moreover, the responses of different species of lady beetles and other predators should be investigated to better understand whether the patterns we observed apply to predators in general or are specific responses of *H. convergens*. Expanding our understanding of natural enemies’ responses to cotton treated with plant-associated fungi will improve our ability to utilize fungal treatments as part of IPM strategies.

## Figures and Tables

**Figure 1 insects-13-00157-f001:**
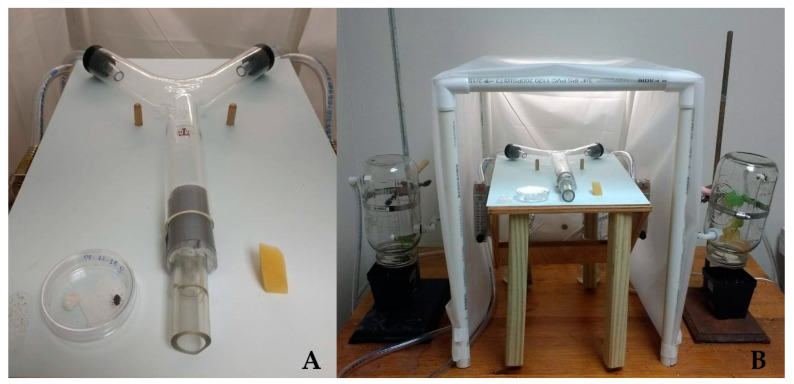
Y-shaped glass tube with the acclimation chamber (**A**) and the entire olfactometer setup showing the plant chambers (**B**).

**Figure 2 insects-13-00157-f002:**
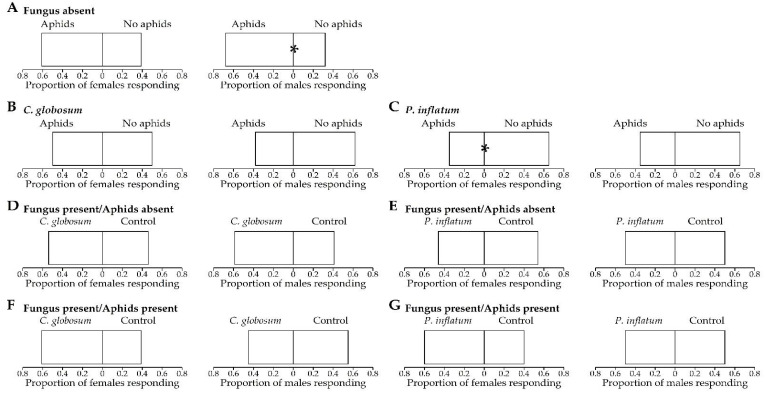
Proportion of *Hippodamia convergens* females and males responding to untreated control and fungal-treated (*Chaetomium globosum* and *Phialemonium inflatum*) cotton plants in a dual-choice Y-tube olfactometer. (**A**–**C**) Untreated plants, *C. globosum,* and *P. inflatum* treated plants with aphids vs. no aphids, respectively. (**D**,**E**) Fungal-treated plants vs. untreated plants, both without aphids. (**F**,**G**) Fungal-treated plants vs. untreated plants, both with aphids. Each individual had 300 s (five minutes) to make a choice from a total of 600 s, and the beetles that did not respond were not included in the analysis. * *p* < 0.05 (Pearson’s chi-squared test).

**Figure 3 insects-13-00157-f003:**
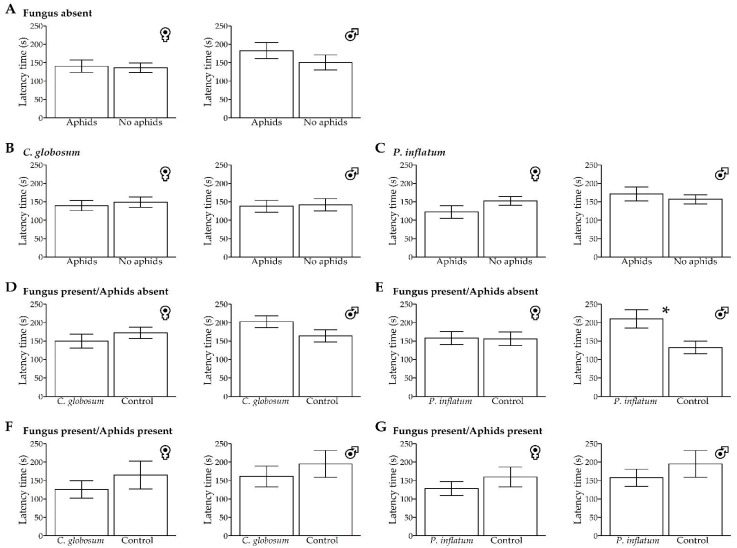
Means (±SE) of *Hippodamia convergens* female and male latency (seconds) to make a choice between olfactory stimuli emitted from untreated control and fungal-treated (*Chaetomium globosum* and *Phialemonium inflatum*) cotton plants in a dual-choice Y-tube olfactometer. (**A**–**C**) Untreated plants, *C. globosum,* and *P. inflatum* treated plants with aphids vs. no aphids, respectively. (**D**,**E**) Fungal-treated plants vs. untreated plants, both without aphids. (**F**,**G**) Fungal-treated plants vs. untreated plants, both with aphids. Each individual had 300 s (five minutes) to make a choice from a total of 600 s, and the beetles that did not respond were not included in the analysis. * *p* < 0.05 (Welch’s two-sample *t*-test).

**Figure 4 insects-13-00157-f004:**
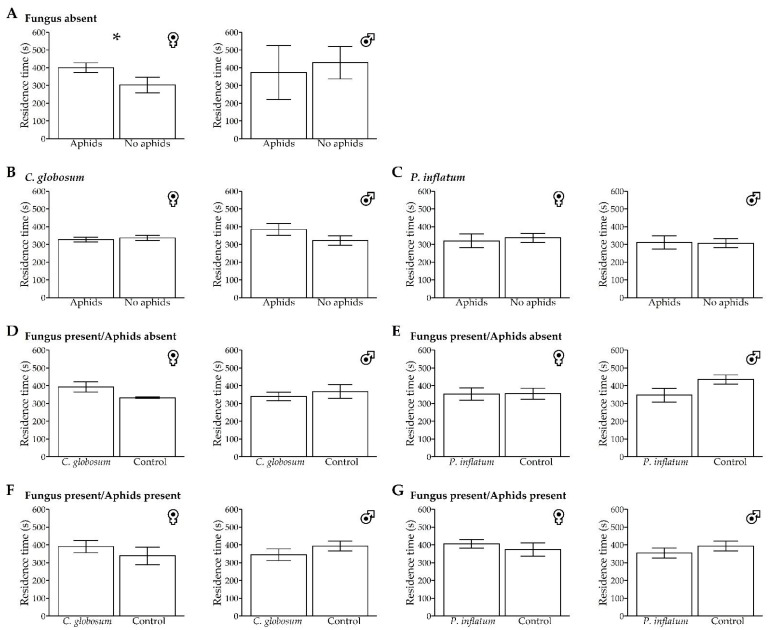
Means (±SE) of *Hippodamia convergens* female and male residence times (seconds) associated with olfactory stimuli emitted from untreated control and fungal-treated (*Chaetomium globosum* and *Phialemonium inflatum*) cotton plants in a dual-choice Y-tube olfactometer. (**A**–**C**) Untreated plants, *C. globosum,* and *P. inflatum* treated plants with aphids vs. no aphids, respectively. (**D**,**E**) Fungal-treated plants vs. untreated plants, both without aphids. (**F**,**G**) Fungal-treated plants vs. untreated plants, both with aphids. Each individual had 600 s (10 min) to stay in the Y-tube arm, and the beetles that did not respond were not included in the analysis. * *p* < 0.05 (Welch’s two-sample *t*-test).

**Table 1 insects-13-00157-t001:** Statistical analyses of the first choice, latency, and residence time in seconds for female and male *Hippodamia convergens*. Tests were conducted in a Y-tube olfactometer providing individuals with a choice between stimuli emitted by fungal-treated or untreated cotton plants in the presence or absence of aphids. Sample sizes for each comparison were N = 60 for each sex. * *p* ≤ 0.05.

	Pearson’s Chi-Squared Test First Choice				Welch’s Two-Sample *t*-Test Latency Time				Welch’s Two-Sample *t*-Test Residence Time				Nonresponding Individuals	
	♂		♀		♂		♀		♂		♀		♂	♀
	χ^2^	*p*	χ^2^	*p*	*t*	*p*	*t*	*p*	*t*	*p*	*t*	*p*		
**Fungus absent**														
Aphids vs. no aphids	4.57	0.03 *	1.77	0.18	1.18	0.26	1.44	0.17	−1.98	0.07	2.13	0.05 *	23	24
** *C. globosum* **														
Aphids vs. no aphids	2.08	0.15	0	1	0.78	0.45	−0.45	0.66	1.04	0.32	−0.23	0.82	21	16
** *P. inflatum* **														
Aphids vs. no aphids	3.27	0.07	4.92	0.02 *	0.74	0.48	−1.44	0.17	−0.03	0.98	−0.39	0.69	23	8
**Fungus present/Aphids absent**														
*C. globosum* vs. control	1.12	0.29	0.23	0.63	1.7	0.1	−1.38	0.19	−0.6	0.55	1.14	0.27	28	21
*P. inflatum* vs. control	0	1	0.23	0.63	2.56	**0.02 ***	−0.11	0.91	−1.9	0.07	−0.01	0.1	38	21
**Fungus present/Aphids present**														
*C. globosum* vs. control	0.53	0.47	2.27	0.13	−0.79	0.43	−1.13	0.28	−1.15	0.26	1.35	0.20	8	11
*P. inflatum* vs. control	0	1	2.08	0.15	−1.07	0.29	−0.97	0.35	−1.01	0.32	1.17	0.25	8	12

## Data Availability

The data presented in this study are available on request from the corresponding author.

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
