# Peer review of "Olfactometer Responses of Convergent Lady Beetles Hippodamia convergens (Coleoptera: Coccinellidae) to Odor Cues from Aphid-Infested Cotton Plants Treated with Plant-Associated Fungi"

_insects, 2022, doi:10.3390/insects13020157_

Round 1

Reviewer 1 Report

The manuscript describes the effects of colonization by endophytic fungi on the attraction of convergent ladybeetle adults to cotton plants infested or not with cotton aphids. Attraction to potential volatiles released from cotton plants under various fungal colonization and aphid infestation was measured using a Y-tube olfactometer. The manuscript was well-written and represents a considerable amount of work.  However, the results are inconclusive.  I believe this is because the Y-tube bioassay was not discriminatory enough. It should have very clearly shown a strong preference by ladybeetles for aphid infested cotton over uninfested cotton, based on previous cited research "H. convergens have previously been shown to be strongly attracted to the odor emitted by plants infested with aphids [68]”. Since your bioassay didn’t allow you to find a significance difference in attraction to volatiles from aphid-infested plants, it would be challenging to see a perhaps minor increase or decrease in attraction due to colonization by the two fungi used in your manipulative studies. The authors pose some great ideas at the end of the discussion that could be followed up on, but solid indication that the bioassay works and that cotton volatile emission is indeed affected by aphid infestation and fungal colonization are necessary first.

I pose several questions below.

  • What was the infestation rate of the aphids on the plants? What was the starting density and the density at the time of use in the assay? Perhaps there weren’t enough aphids to cause changes in the volatile release profile that would affect beetle behavior. The authors state that colonization by some fungi reduces aphid reproduction. “However, this same strain of fungi has previously been shown to reduce aphid population growth on cotton when applied in the same manner (5)]”. Were there any differences in final aphid numbers on the fungus-colonized and uncolonized plants that could have influenced amounts of volatiles released?
  • Most people doing Y-tube assays consider an insect to have made a choice after it has moved half-way up an arm, rather than simply entering the arm. Why did you choose to use first choice as your variable?

Minor points

  • Metromix soil was described as from the Borlaug institute. If it is commercially available, it should be cited as manufacturer, city, state.
  • Experiments were done under natural light conditions – this will vary during the season. Photoperiod during the course of the experiments should be described.
  • Mason jar – this might not be known to international readers. Describe as a glass jar.
  • “The first-choice responses from convergens wild and commercial females and males were not significantly different (F3, 4 = 3.576, p = 0.125” This seems like a very small error df. Is this correct?

Reviewer 2 Report

This paper report a behavioral study carried out solely in laboratory with the use of Y-shaped olfactometer to assess the influence of symbiotic fungi in the trophic interaction “cotton plants-aphid-predator”. Overall the paper is without praise or blame: fairly written, the experiment involves an adequate number of replicates and the data are analyzed in depth. However, no chemical analysis from of the VOCs of the different treated and untreated plants were carried out. Introduction is too brief and does not give a proper background of the context of this work, in addition objectives are not well pointed out. Furthermore, a methodological main issue related on the structure of the jars linked with the olfactometer must be better explained: the use of plants with only aerial parts inside the olfactometer is interesting. However, is not clear how the jar were closed at the bottom sealing around the plant stem, this can be critical to avoid the air entrance from outside without damaging the plant.

In abstract and simple summary, the name of the species is not in italic. In both abstract and simple summary is not clear if the there is a difference or not in the response to the different stimuli tested between the two sexes.

The discussion is well written but is very speculative, in consideration that this was a preliminary-laboratory study and that the conditions of the experiment reported are very different from the agricultural ecosystems, with their very complex stimuli background.

Throughout the manuscript verify that for each species mentioned for the first time is reported the order and the family.

Lines 9-12, The first three sentences of the simple summary are not linked together. Overall the simple summary is totally to rewrite, as is very fragmentary and badly written in general.

Line 56,  “The cotton aphid is a serious global agricultural pest.” Which aphid species?

Lines 56-62, these two sentence are not connected.

Lines 68-76, this part should be implemented.

In section 2.2. Insect rearing and experimental design, there is almost nothing about the “insect rearing”. Give some more detail about where the insects were kept (climatic cell?), and how they were fed until the experiment.

Lines 124-127, this part is rather unclear and is not introduced in the objectives.

Section 2.4 avoid to specify the function in R

Figure 1B, I’m a bit concerned about the position of the plants that are actually not totally inside the glass jars: how was the bottom of the glass jar closed, separating only the upper part of the plant? How can the authors avoid the entering of air from the outside? Please explain in detail.

Section 3.1. Wild and commercial H. convergens responses. Was this one of the objectives?

In Figure 2 caption is reported that beetles had 300 seconds to respond, while in MM above is stated that the beetles had 600 seconds to respond. Please advise.

Author Response

Thanks!

Round 2

Reviewer 2 Report

In general, I think that the paper is greatly improved from its old version and it can now be considered for publication after solving some minor points. My main concern is the fact that, once this study was conducted only using a two-choice olfactometer, the title can be misleading. I think that the fact that this is an olfactometer study should be pointed out in the title, that (in my humble opinion) at the moment is promising too much.

I would consider a title like “Olfactometer response of the convergent lady beetles Hippodamia convergens (Coleoptera; Coccinellidae) to odor cues from aphid-infested cotton plants treated with plant-associated fungi”

Few other minor issues:

Line 13, point out that this is a natural enemy of the aphid

Line 19, unclear sentence.

Line 112, add “USA”

Line 124, add “USA”

Line 260, once is at the beginning of a sentence I would not abbreviate the species name.
